# Production of Polyhydroxyalkanoates and Extracellular Products Using *Pseudomonas Corrugata* and *P. Mediterranea*: A Review

**DOI:** 10.3390/bioengineering6040105

**Published:** 2019-11-14

**Authors:** Grazia Licciardello, Antonino F. Catara, Vittoria Catara

**Affiliations:** 1Consiglio per la Ricerca in agricoltura e l’analisi dell’Economia Agraria-Centro di ricerca Olivicoltura, Frutticoltura e Agrumicoltura (CREA), Corso Savoia 190, 95024 Acireale, Italy; 2Formerly, Science and Technologies Park of Sicily, ZI Blocco Palma I, Via V. Lancia 57, 95121 Catania, Italy; antoninocatara@virgilio.it; 3Dipartimento di Agricoltura, Alimentazione e Ambiente, Università degli studi di Catania, Via Santa Sofia 100, 95130 Catania, Italy; vcatara@unict.it

**Keywords:** medium-chain-length polyhydroxyalkanoate (mcl-PHA), alginate, biosurfactants, biopolymer, *Pseudomonas*, blends, film

## Abstract

Some strains of *Pseudomonas corrugata* (*Pco*) and *P. mediterranea* (*Pme*) efficiently synthesize medium-chain-length polyhydroxyalkanoates elastomers (mcl-PHA) and extracellular products on related and unrelated carbon sources. Yield and composition are dependent on the strain, carbon source, fermentation process, and any additives. Selected *Pco* strains produce amorphous and sticky mcl-PHA, whereas strains of *Pme* produce, on high grade and partially refined biodiesel glycerol, a distinctive filmable PHA, very different from the conventional microbial mcl-PHA, suitable for making blends with polylactide acid. However, the yields still need to be improved and production costs lowered. An integrated process has been developed to recover intracellular mcl-PHA and extracellular bioactive molecules. Transcriptional regulation studies during PHA production contribute to understanding the metabolic potential of *Pco* and *Pme* strains. Data available suggest that *pha* biosynthesis genes and their regulations will be helpful to develop new, integrated strategies for cost-effective production.

## 1. Introduction

Polyhydroxyalkanoates (PHAs) are microbial polyesters synthesized by both Gram-negative and Gram-positive eubacteria, and an increasing number of archaea isolated from environmentally extreme habitats, to increase their survival and competition in environments where carbon and energy sources are limited, such as soil and rhizosphere [1,2,3]. 

Based on their repeat unit composition, the up to 150 different PHA structures identified so far [4] are classified mainly in two distinct groups: (i) short chain length (scl) PHAs where the repeat units are hydroxy fatty acids (HFAs) of 3–5 carbon chain length (C3–C5); and (ii) medium chain length (mcl) PHAs with repeat units of C6–14. In general, scl-PHAs are crystalline polymers with a fragile, rigid structure, whereas mcl-PHAs are amorphous thermoplastics, which have various degrees of crystallinity as well as elastomeric and adhesive properties [5]. Less common and least studied are long chain length (lcl) PHAs, constituted of monomers with more than 14 carbon atoms.

Thanks to two metabolic pathways based on the degradation of aliphatic carbon sources or *de novo* synthesis of fatty acids from unrelated carbon sources, *Pseudomonas* species included in the rRNA homology group I are among the most important producers of PHA [6,7,8,9]. Historically, fatty acids have been the preferred substrate for the microbial synthesis of mcl-PHA. Glucose, gluconate or ethanol, as well soy molasses [10], biodiesel co-product stream [11] and glycerol [12,13], have been successfully used. The fatty acyl composition of the substrate reflects the repeat unit composition of biopolymers [14]. Biodiesel glycerol has been recognized as a suitable and cost attractive substrate for PHA production, and therefore constitutes the main focus of this review [15,16].

Medium chain length-PHAs in *Pseudomonas* bacteria were first detected in *P. oleovorans* [17] and later in a variety of *Pseudomonas* [10]. *Pseudomonas*-PHAs are biodegradable, non-toxic and biocompatible and can be produced using a wide range of carbon sources. In fact, there has been considerable research exploring their potential in medical devices, foods, agriculture and consumer products [18,19]. Their elastic and flexibility properties improve the processability and mechanical properties of blends with other biodegradable polymers [20,21,22]. Of the various species tested worldwide, *P. aeruginosa*, *P. putida*, *P. resinovorans, P. mendocina*, and *P. chlororaphis* are the most extensively studied to clarify the metabolic processes of the production of PHA and to enhance the bioconversion efficiency [3,23].

This review focuses on the two taxonomically related Gram-negative rod ubiquitous bacteria, *P. corrugata* and the strictly related *P. mediterranea*, which cause disease on several crop species [24] and can produce an arsenal of secondary metabolites [25]. Among them, are biosurfactants (BSs) [26] as well as poly-mannuronic acid alginate [27], bioactive cyclic lipopeptides (CLPs), such as cormycin A and corpeptins [28,29,30], and a lipopeptide siderophore, corrugatin [31].

These bacteria produce different cellular mcl-PHAs and extracellular products, on waste fried edible oils, biodiesel glycerol and high-grade glycerol [13,32,33]. Selected *P. corrugata* strains produce intracellular mcl-PHA with a molecular weight of 120–150 kDa on waste edible oils, whereas strains of *P. mediterranea* generate a distinctive filmable PHA around 55–65 kDa on high-grade and partially refined biodiesel glycerol. Extracellular products, such as biosurfactants, exopolysaccharides (EPS, mostly alginate) and bioactive molecules, accumulate in the supernatant during the bioconversion process. Genome analysis of nine *P. corrugata* and *P. mediterranea* strains has helped to develop molecular and genetic investigations to enhance productivity [25].

## 2. Production of mcl-PHA and Extracellular Products

The first strain of *P. corrugata* investigated for its capacity to convert triacylglycerols to produce mcl-PHA was strain 388 [7,14,34]. The positive results led to the screening of different carbon sources of 56 strains of *P. corrugata* and 21 strains of its closely related *P. mediterranea* [9]. Flask-scale tests, carried out on related and unrelated carbon sources, have been reported [9,13,32,33,34].

Subsequently, some strains of *P. corrugata* producing lipase have been reported as being able to bioconvert waste exhausted fried edible oils, from a licensed collector, in mcl-PHAs [9,32]. One of these strains, namely *P. corrugata A1* (DSM 18227) (hereafter *Pco A1*), obtained through culturing *P. corrugata* CFBP5454 in E* medium with triolein, helped to patent a fermentation process validated on a 5000 L fermenter [35]. This increased the productivity of the process from 2.90 g/L up to 26 g/L of dry cell weight with 38% of PHA [32,35]. 

To overcome the variable composition of licensed exhausted edible oils and the difficulty in collecting adequate stocks for industrial production, several sources of glycerol have been extensively tested to screen many strains of *P. corrugata* and *P. mediterranea*. Three of them, *Pco 388*, *Pco A1* and *P. mediterranea 9.1* (deposited as CFBP5447, hereafter *Pme 9.1*), have been selected to study the bioconversion processes exploiting commercial high-grade glycerol (≥99%, pH 7) and biodiesel glycerol obtained from the transesterification of rapeseed oils (*Brassica carenata* and *B. napus*). Crude biodiesel (15% glycerol), oil free (40%) and partially refined glycerol (87.5%) performed differently from commercial high grade glycerol in terms of yield, composition and properties of the mcl-PHA and extracellular products. It has also been highlighted that some apparently small differences in the carbon sources may have a large impact, and that the genetic and metabolic system of the strain are key to the bioconversion process [11,12,13,33].

*P. mediterranea 9.1* reached a production of 2.93 g/L of mcl-PHA on 2% crude biodiesel glycerol with a PHA/cell dry weight ratio >60%, whereas on high-grade glycerol it yielded 0.81 g/L (Table 1) [33]. In parallel tests, *P. corrugata* A.1 produced 1.8 g/L of PHA with 51.5% in cultivation for 72 h [33]. The productivity of batch fermentations on E* medium with the addition of 1% or 2% of glycerol showed only minor differences, but decreased yields and mass molecular weight (Mw) were observed when 5% of glycerol was added. The same results were reported by Ashby et al. [11], in the case of Mw of PHB produced by *P. oleovorans* and mcl-PHAs accumulated by *P. corrugata 388*.

Another flask-scale experiment, carried out with *Pme 9.1* growing on a medium containing 2% refined glycerol, yielded 3.3 g/L in cell dry weight (CDW) after cultivation for 48 h, with a PHA/CDW ratio close to 18% (Table 1) [13]. No significant changes were observed after 60 and 72 h of cultivation. Soxhlet extraction of biomass with acetone produced 0.75 g/L of a thin opalescent film of crude mcl-PHA. Parallel fermentation carried out with partially refined glycerol (87.5%) obtained from the esterification of *B. napus* oil produced 3.1 g/L of biomass and 0.5 g/L of raw PHA (PHA/CDW = 16.5%) [13]. 

Besides the different conversion efficiencies, other chemical and technological properties of the PHA were even more relevant. Regardless of the carbon source, both strains of *P. corrugata* (*388* and *A1*) produced very similar mcl-PHA elastomers, whereas PHA obtained from *P. mediterranea* 9.1 grown on refined biodiesel glycerol, generated a transparent filmable polymer with a low molecular weight (56,000 Da) and very distinctive characteristics (Figure 1) [13]. 

The extracellular biosurfactants released by these strains during the bioconversion process showed their dependence on the carbon sources, and the highest yields were reached much later than the PHA. *Pme 9.1* grown on crude glycerol (15%) obtained from *Brassica* spp. seed oil, was able to recover up to 14 g/L of surfactants with E24 (emulsification index) 54%, via chloroform:methanol (2:1) [36]. The highest accumulation occurred after 96–144 h. At the early stationary phase (48 h) *P. mediterranea 9.1* yielded 6.9 g/L of partially purified EPS, 17-fold higher than in *Pco A1* (0.39 g/L). PHA production was slightly higher in *Pco A1* than in *Pme 9.1* (respectively 0.92 g/L and 0.52 g/L) [37].

## 3. Conversion Process and Recovery

In order to establish standard and suitable protocols to scale up the production of PHAs, many strategies have been investigated using batch and fed-batch processes in flasks and low-medium volume fermenters (3–30 L). Fed-batch fermentation has always been shown to be more productive than the batch mode, as reported for *Cupriavidus* sp. [40].

A fed-batch cultivation has also been used in a process of glycerol conversion by growing *P. mediterranea 9.1* in a substrate with 2% glycerol. The cultivation was conducted in a 30 L bioreactor, 30 °C, pH 7.0, and dissolved oxygen maintained at 20% saturation, using E* medium (pH 7.0) containing 5.8 g/L K_2_HPO_4_, 3.7 g/L KH_2_PO_4_, 10 mL/L MgSO_4_ 0.1 M, supplemented with 1 mL/L of a microelement solution, with the addition of 2% glycerol (1% at the start, and 1% after 24 h) [13,33].

The biomass obtained has been routinely harvested by centrifugation, washed with saline solution and lyophilized. The extraction of the PHA using acetone [41] in an automatic Soxhlet was found to be more effective than chloroform extraction and less impactful for the environment. Treatments with mild alkaline solution [42] or maceration, attempted considering the potential use of mcl-PHA in the biomedical field, have yielded a lower recovery of products.

The analysis of PHA composition was carried out by gas chromatography/mass spectrometry (GC/MS) of the 3-hydroxymethylesters, after the removal of all the residual free glycerol [13]. Overall, different approaches have been evaluated to reduce the very high production costs by increasing the yield or by recovering both the PHA and extracellular products simultaneously from the fermentation process. The addition of either meat or yeast extracts at 0.1% to crude glycerol or glucose eliminated the prolonged lag-phase (5–12 h) [43].

Rizzo et al. [44] showed that adding 5 mM glutamine as a co-feeder significantly increased the biomass and PHA production, inducing the early expression of *phaC1* and *phaC2* genes. This was due to the improvement in the specific growth rate and cell metabolic activity, and to the enhanced uptake of the unrelated (glycerol and glucose) and related (sodium octanoate) carbon sources.

An integrated process for the bioconversion of crude biodiesel glycerol to simultaneously produce biosurfactants and PHAs by *Pme 9.1*, has also been established by applying a mathematical mechanistic model to define nutritional requirements, as well as pH and temperature, which mutually influenced PHA and BSs production within a narrow range of variation [43]. Surface response methodology analysis showed that, after 72 h, up to 1.1 g/L of crude PHA and 0.72 g/L of biosurfactants were recovered. On the other hand, the respective best single yields were obtained after 48 h for PHA (60% of CDW) and 96 h for BSs (0.8 g/L) [43].

## 4. Composition and Technological Properties of mcl-PHA 

GC/MS profiles of mcl-PHA were largely affected by the carbon source and bacteria species. PHAs obtained on waste food oils have been found to be very different from those obtained on glycerol, and different types of glycerol produced different mcl-PHAs [13,32].

GC/MS chromatograms of mcl-PHAs obtained by *Pco A1* and *Pme 9.1* on crude glycerol revealed similar profiles and technological properties, whereas substantial differences were observed with respect to those obtained on partially refined biodiesel glycerol and high-grade glycerol. They showed monomeric units of side chains from C12 to C19 in length on crude glycerol (15% glycerol), and from C5 to C16 on refined glycerol (≥99%) (Table 2) [33,45]. Interestingly, mcl-PHA produced by *Pme 9.1* on high-grade glycerol was less sticky and produced a thin film (Figure 1A) [33]. These properties have been shown to be associated with differences at transcriptomic level [37], and in the genetic organization of *pha* gene locus which affects *pha* polymerase gene expression, PHA composition, and granule morphology [39].

Other experiments on *Pme 9.1* have been conducted in Erlenmeyer flasks containing 500 mL volumes of E* medium (pH 7.0) with 2% high grade glycerol or a partially refined glycerol (87.5%) obtained from a biodiesel process of *Brassica napus* [13]. The polyesters obtained on high grade commercial glycerol highlighted a structure composed of six monomers, indicative of elastic and flexibility properties: 3-hydroxyhexanoate (C6), 3-hydroxyoctanoate (C8), 3-hydroxydeca-noate (C10), 3-hydroxydodecanoate (C12), cis 3-hydroxydodec-5-enoate (C12:1∆^5^), and cis 3-hydroxydodec-6-enoate (C12:1∆^6^). The molecular weight (Mw) was 55,480 Da and polydispersity index (PDI = Mw/Mn) was 1.34 (Table 2). On the other hand, PHA obtained from glycerol 87.5% had a small variation in monomeric composition, a Mw of 63,200 Da, and a PDI of 1.38. Tsuge et al. [46] also observed that a higher glycerol concentration induced a considerable reduction in the molecular mass of PHA, caused by a termination of the PhaC polymerization activity. The NMR spectra and MALDI-TOF data were almost identical regardless of the glycerol grade, but different in intensity. The degradation temperature started at 230 °C, higher than the melting temperatures, with a volatilization rate of about −40%/min. 

Drop casting a toluene solution of polymers in Petri dishes resulted in quite different films, depending on the carbon source used to produce the PHA (Figure 1B). The PHA obtained on high-grade glycerol produced an optically transparent film with a UV–vis absorption spectrum that was above the 800–350 nm range, comparable to the polyester film used for laser printer transparency.

The mechanical proprieties (tensile strength, Young’s modulus and elongation at break of both PHAs) were not substantially affected by the different purities of the glycerol grade (87.5% and ≥99%). All these characteristics make the PHA obtained from *P. mediterranea 9.1* on glycerol quite different to most mcl-PHAs produced from bacteria of the same phylogenetic group.

## 5. Evaluation of Mixed Blends and Coatings 

The distinctive characteristics of a mcl-PHA obtained from *Pme 9.1* grown on glycerol led to the investigation of the processability of blends with polylactide acid (PLA) to improve the mechanical and gas/vapors barrier properties of PLA [48]. Rheological tests indicated a significant increase in the elongation at break, while the elastic modulus was significantly lower only at higher contents of PHA. This suggests that the PHA macromolecules exert both a plasticization and lubricant action, which enable the PLA macromolecules subjected to solid deformation to slide more efficiently [48].

Preliminary investigation of blends of polyhydroxybutyrate (PHB) and a glycerol mcl-PHA obtained from *B. napus* oil showed an increase in crystallization temperature and a small increase in elongation at break, but at low concentrations of PHA (5%) the blend revealed some spaces between the two polymers.

Blends of mcl-PHA obtained from *Pco A1* on exhausted edible oils with Mater-Bi ZI01U/C polymers have poorly improved the processability of blends prepared by compression [22]. Soil mulching tests of paper sheets coated with blends based on PHA suggested some positive effects of coating. However, the expensive costs, as well the difficulty to obtain standardized exhausted edible oils as a carbon source, have discouraged further research [49,50].

## 6. *P. corrugata* and *P. mediterranea* PHA Locus 

Genomic studies which investigated potential correlations between the phenotype and genotype of *Pco 388*, *Pco A1* and *Pme 9.1* have shown that, similarly to other *Pseudomonas*, the three strains have a class II PHA genetic system consisting of two synthase genes (*phaC1*, *phaC2*), separated by a gene coding for the depolymerization of PHA (*phaZ*) [8,9,23]. This genetic system allows *Pseudomonas* strains to utilize medium-chain-length (mcl) monomers (C6–C14), whereas class I, III and IV systems polymerize short-chain-length (scl) monomers (C3–C5) [6].

Sequence analysis of the *pha* locus revealed that the strains *Pco A1* (AY910767), *Pmed 9.1* (AY910768) and *Pco 388* (EF067339) share a high homology at nucleotide (93–95%) and amino acid levels (96–98%) [39,51,52]. An additional 121 bp in the *phaC1–phaZ* intergenic region containing a predicted strong hairpin structure were present in both strains *388* and *A1* of *P. corrugata*, but not in *P. mediterranea* [39]. According to the authors, in *Pco A1* and *388* strains this additional sequence likely acts as a rho-independent terminator for the transcriptional terminator of *phaC1*, which would appear to be responsible for the slight variation in the PHA composition and granule organization [39]. 

Subsequently, genome analysis of strains *Pco A1* (ATKI01000000) and *Pme 9.1* (AUPB01000000) enabled the six genes of the entire *pha* locus to be studied (*phaC1*, *phaZ*, *phaC2*, *phaD*, *PhaF*, *PhaI*) [53,54]. Genome mining also identified gene coding for enzymes involved in β-oxidation (*fad*), fatty acid *de novo* synthesis (*fab*), and mcl-PHA precursor availability (*phaG* and *phaJ*) [53,54]. Pfam search domain and Blastp analysis on the *glp* operon, responsible for glycerol catabolism, revealed that both *P. corrugata* and *P. mediterranea* lack the *glpF* gene, coding for the glycerol uptake facilitator protein [53]. This condition had already been verified in all the *Pco* and *Pme* strains sequenced, which explains the prolonged lag growth phase observed during *P. mediterranea 9.1* growth with glycerol as the sole carbon source [25,55].

## 7. Transcriptional Regulation during PHA Production

To improve the knowledge about PHA biosynthesis genes and their regulations, helpful to increase mcl-PHAs production and also to obtain new, tailor-made polymers [6], regulatory mechanisms during PHA accumulation have been investigated in *Pco A1*, *Pco 388* and *Pme 9.1* through different gene expression studies and full transcriptome analysis.

### 7.1. Expression of phaC1 and phaC2 under Different Carbon Sources

Preliminary studies on the transcriptional levels of *phaC1* and *phaC2* genes in *Pco 388* and *Pco A1* showed an up-regulation of *phaC1* in cultures with oleic acid as the sole carbon source (Table 3) [47]. On the other hand, both *phaC1* and *phaC2* were induced in cultures with glucose or sodium octanoate [47]. The significant correlation between PHA production and *phaC1/phaC2* expression suggested at least two distinct networks for the regulation of the two PHA polymerases genes, and that a putative promoter(s) is likely present upstream of *phaC2*. In addition, the lack of polycistronic transcripts under any culture conditions indicated that *phaC1* and *phaC2* were not co-transcribed (Table 3) [47].

Parallel studies showed that, in *Pco 388*, the *phaC1–phaZ* intergenic region plays an important but unclear role in the regulation of the carbon source-dependent expression of *phaC1* and *phaC2* genes [39]. Derivative mutants XI 32-1 and XI 32-4 of this strain (obtained by replacing the *phaC1–phaZ* intergenic region with a kanamycin resistance gene), showed a significant increase in *phaC1* and *phaC2* expression when grown for 48 h with oleic acid, but not with glucose. In addition, the wild type strain produced only a few large PHA inclusion bodies when grown with oleic acid, whereas the mutants showed numerous smaller PHA granules that line the periphery of the cells, as result of phasin activities [3,39]. A high content of the monounsaturated 3-hydroxydodecanoate as a repeat unit monomer was observed in the PHA of the mutant strains [39].

Diversely, the study of the promoter activity of *pha* genes in *Pme 9.1* grown on high-grade glycerol, revealed that the upstream regions of *phaC1* (PC1) and *phaI* genes (PI) are the most active [56]. PC1 is responsible for the *phaC1ZC2D* polycistronic unit transcription (Table 3). On the other hand, PI regulates the *phaIF* operons, as confirmed by the presence of three and two putative rho independent terminators, respectively located downstream of *phaD* and *phaF* [56]. In turn, PI and PC1 are controlled by PhaD, which acts as a transcriptional activator, as shown by the reduced promoter activities in the *phaD*- mutant [55]. Similar results were observed in *P. putida* KT2442 [23,57].

### 7.2. Transcriptome Analysis on Glycerol-Grown Strains

The transcriptional profiles of *Pco A1* and *Pme 9.1* growing on a substrate with 2% of high-grade glycerol under inorganic nutrient-limited conditions were investigated at the early stationary phase of the bioconversion into mcl-PHAs [37]. RNA-seq analysis revealed that in *P. mediterranea*, 175 genes were significantly upregulated and 217 downregulated, compared to *P. corrugata.* The genes responsible for stress response, central and peripheral metabolic routes and transcription factors involved in mcl-PHA biosynthesis, made up 39% of the genes differently transcribed by the two bacteria. Nonetheless, among the genes directly involved in PHA biosynthesis, slight differences were observed only in *phaZ* depolymerase and *phaG* transacylase genes (Table 3). Weak differences occurred in the expression levels of genes that are crucial for glycerol catabolism and pyruvate metabolism, transcriptionally downregulated, and fatty acid *de novo* biosynthesis pathways. 

Interestingly, a significantly increased expression of 21 genes involved in alginate exopolysaccharide production was observed in *Pme 9.1* compared to *Pco A1* (Table 3), related to a 17-fold higher production of EPS (6.9 g/L compared to 0.39 g/L). A simultaneous production of PHA and alginate has been reported in some *P. mendocina* strains [58,59]. The increased EPS production, associated with the different transcriptome profiling between the two bacteria, suggests competition for the acetyl-CoA precursor amongst PHA and alginate metabolic pathways. Further studies on *P. corrugata A1* showed that regulation of alginate production is controlled by quorum sensing and the RfiA regulator [60].

## 8. Genetically Modified Bacteria to Improve the Production of mcl-PHAs

Two different approaches have been made to evaluate the feasibility of improving the efficiency of the conversion by using genetically modified-*Escherichia coli* and *P. mediterranea*.

Cloning of *pha* synthases genes of *Pco A.1* and *Pme 9.1* in *Escherichia coli,* a well-known organism in the research on PHA biosynthesis, yielded 2%–4% of PHA/CDW on sodium decanoate [61]. 

In a second approach, additional copies of *phaC1*, *phaG* and *phaI* genes, cloned in two plasmids under the control of strong promoters, were transferred into *Pme 9.1* [38]. When grown on high-grade glycerol, the modified *Pme VVC1GI* showed a higher cell fluorescence than WT, due to the presence of larger granules (Figure 2). It also showed a 40% increase in the cellular accumulation of crude PHA and a better PHA/CDW ratio (1.4 g/L, 38.8%), whereas the WT strain produced 1 g/L of PHA (23%) (Table 1). Biomass yield was 3.6 g/L in *VVC1GI* and 4.2 g/L in WT [38]. GC/MS analysis showed that the PHA structure was composed of six monomers, like PHA produced by the WT, although there were some differences in the ratio between C12 and C12:1 (12.8:11.8 in strain *VVC1GI* and 6:12 in Wt) (Table 2).

## 9. Conclusions

Despite the fact that the PHAs market is still very small, the worldwide focus on the development of bio-polymers highlights that one day polyhydroxyalkanoates will replace some petroleum-based plastics. *Pseudomonas* species are a potential cell factory for their production [3]. Some strains of *P. corrugata* and its related *P. mediterranea* are able to convert different carbon sources and provide interesting mcl-PHA and co-products. They are naturally present in the soil and produce valuable extracellular co-products [37,43]. 

A manageable mcl-PHA film, unlike other mcl-PHAs reported to date, can be obtained by *P. mediterranea 9.1*. Although the yields are currently not particularly viable, their distinctive characteristics suggest a potential application as a softener in (bio) polymeric blends, for food packaging or medical devices. The unsaturated double bonds in the side chains could be used to enhance its properties and/or to help extend its applications to other biomaterials for food packaging or biomedicine [13]. 

*P. mediterranea 9.1* also produces high-quality extracellular products (above all alginate) on a proper medium, which is very promising for high-level applications and which may orient further investigation towards an efficient co-production of cellular mcl-PHA and extracellular biosurfactants, EPS and other bioactive molecules [43]. These results and those available for other systems highlight the potential of such integrated microbial conversion processes [62]. Further top strategies are required to find solutions for the industrial production of such compounds and new ones.

Pioneering work on other PHA producers *Pseudomonas*, such as on expanding the number of inexpensive carbon sources [63], increasing the productivity [64,65] or making the PHA deposition extracellular [66], highlight the potential for successful future investments in this sector. In the meantime, given that robust strains are needed to reduce the high production costs, using genetic engineering and metabolic studies on these two bacteria should focus on developing over-producer strains of mcl-PHA, as well as the co-production of other valuable products, such as EPS and biosurfactants. 

## Figures and Tables

**Figure 1 bioengineering-06-00105-f001:**
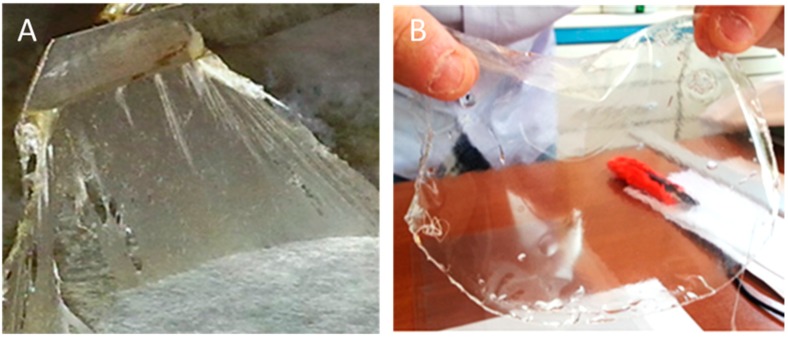
Crude PHA film (**A**) and transparent PHA film obtained after floating a toluene solution on a water surface (**B**) achieved from *Pseudomonas mediterranea* 9.1 using refined glycerol as carbon source (Figure 1B courtesy of Copyright Elsevier from [13]).

**Figure 2 bioengineering-06-00105-f002:**
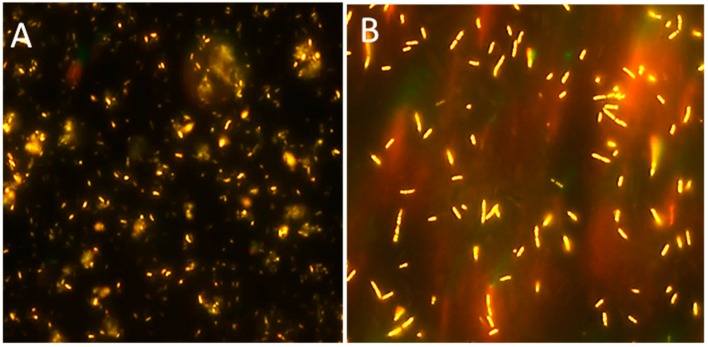
Fluorescent granules of PHA after Nile red-staining of *Pseudomonas mediterranea 9.1* wild-type strain (**A**) and *VVC1GI* recombinant strain (**B**) grown on high-grade glycerol (≥99%) as carbon source and limited nitrogen condition after 66 h of incubation.

**Table 1 bioengineering-06-00105-t001:** Cell dry weight and raw PHA percentage obtained through bioconversion of different carbon sources by selected strains of *Pseudomonas mediterranea* and *P. corrugata*.

Carbon Source	Grade	% V:V	Time (h) ^1^	*P. mediterranea* 9.1		*P. corrugata* A1		*P. corrugata* 388		References
				CDW (g/L)	Raw PHA (%)	CDW (g/L)	Raw PHA (%)	CDW (g/L)	Raw PHA (%)	
Glycerol	15%	1	72	3.4	50.2	4.7	50	4	28.5	[33]
	≥99%			3	25.3	3.5	29.4	4.2	18.7	
	15%	2	72	4.8	61.6	3.5	51.5	3.8	33.6	
	≥99%			3.2	26.1	3.4	30.2	3.6	15.7	
	15%	5	72	4.2	38	4.1	48.5	3.2	32.1	
	≥99%			3.3	21.5	4.1	22.1	2.8	14.3	
Glycerol	87.5%	2	48	3.1	16.5					[13]
	≥99%			3.3	18					
Glycerol	≥99%	2	66	2.9	17.9	3.1	29.4			[37]
Glycerol	≥99%	2	66	3.6 ^2^	38.8 ^2^					[38]
Glucose	≥99%	0.5	72					1.5	31.3	[34]
Oleic acid	≥99%							1.6	61.8	
Oleic acid	≥99%	2	72					3.1	24	[39]
Glucose	≥99%		48					1.3	2	

^1^ time of cultivation; ^2^ this specific test was carried out with a modified strain of *P.mediterranea 9.1 VVC1GI.*

**Table 2 bioengineering-06-00105-t002:** Molecular weight and monomer composition of PHAs obtained in different bioconversion processes of different carbon sources by *Pseudomonas corrugata* and *P. mediterranea*.

Strain	Carbon Source	Grade	% V:V	Time (h) ^1^	Mw (kDa)	PDI	Molar Composition (mol %)								Reference
							C6	C8	C10	C12:0	C12:1	C12:0	C14	C14:1	
*Pme 9.1*	Waste fried oil						2	34	44	14			5		Pappalardo et al., unpublished
	Glycerol	80%	2				1	7	71	8	13		1		
	Glycerol	40%	2				1	15	43	11	7		24		
	Glycerol	≥99%	2	48	55.5	1.34	4.2	17.0	60.8	1.1	11.2	5.7	-	-	[13]
		87.5%			63.2	1.38	0.1	9.3	66.6	1.5	14.8	7.7	-	-	
	Glycerol	≥99%	2	66			4	17	60	7	12		0.4		[37]
*Pme* 9.1 VVC1GI	Glycerol	≥99%						0.9	13.5	57.5	12.8	11.8		3.7	[38]
*Pco A1*	Glucose		0.5	72	125.8	2.4	2	14	52	11	17		0.4	3.6	[47]
	Oleic acid				159.0	1.5	10	48	28	8				6	
	Na octanoate				183.2	2.1	11	82	7						
	Glycerol	≥99%	2	66			2	12	53	14	17		5		[37]
*Pco 388*	Oleic acid		0.5	72	735	4.1		47	24.5					16.5	[8]
	Glucose		0.5	72	nd		2	19	56	11			2	9	[34]
	Oleic acid				nd		5	37	33	12			2	12	
	Na octanoate		0.5	168	114	1.8	7	82	11						[47]
	Oleic acid		2	-			5	54	20	5			15		[39]
	Glucose			-			2	28	35	9	14		9		

^1^ time of cultivation.

**Table 3 bioengineering-06-00105-t003:** Gene expression detected in *P. corrugata* and *P. mediterranea* strains during mcl-PHA biosynthesis on different carbon sources.

Bacterial Strain	Carbon Source	% V/V	Time (h)	Detection Method	*PhaC1*	*PhaC2*	*PhaI*	*Alg* Genes	Operon	Reference
*Pco 388*	Oleic acid	2	48	Real-time PCR ^1^	1.2	1.4	nt	nt	nt	[39]
*Pco 388* clone XI 32-1					6.6	4.7	nt	nt	nt	
*Pco 388* clone XI 32-4					7.0	5.4	nt	nt	nt	
*Pco 388*	Glucose	2			5.6	No change	nt	nt	nt	
*Pco 388* clone XI 32-1					6.3	No change	nt	nt	nt	
Pco 388 clone XI 32-4					8.2	No change	nt	nt	nt	
*Pco A1*	Oleic acid	0.5	48 72	Real-time PCR ^1^	6.8 10.5	No change	nt	nt	NO	[47]
*Pco* 388			48 72		2.7 2	No change	nt	nt	NO	
*Pco A1*	Glucose	2	72		6.2	3.5	nt	nt	NO	
*Pco* 388			72		3.8	3	nt	nt	NO	
*Pme 9.1*	Glycerol	2	24 48	β-gal ^2^	420 U 300 U	340 U 400 U	2200 U 7000 U	nt	*PhaC1ZC2D PhaIF*	[56]
*Pme 9.1* VVD (*phaD*-)			24 48	β-gal	140 U 300 U	350 U 400 U	45 U 45 U	nt	*PhaC1ZC2D PhaIF*	
*Pme 9.1*	Glycerol	2	48	RNA-Seq ^3^	No change	No change	No change	5.53–2.32	nt	[37]
*Pco A1*			48	RNA-Seq	No change	No change	No change		nt	

^1^ The relative quantification was performed by comparing ΔCt (i.e., Ct of the 16S rRNA housekeeping gene subtracted to the Ct of the target gene). The ΔCt value of the control sample (time 0) was used as the calibrator and fold-activation was calculated by the expression: 2^−ΔΔCt^. ^2^ β–galactosidase activities detected by transcriptional fusion plasmids for *phaC1*, *phaC2*, and *phaI* promoter regions based on the pMP220 promoter probe vector and expressed as Miller units. ^3^ Pairwise comparison of mRNA levels analysis, using the *Pme 9.1* sample as a reference (log2 fold change ≥ 2 and *p*-value ≤ 0.05).

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
