# Peer review of "Production of Polyhydroxyalkanoates and Extracellular Products Using Pseudomonas Corrugata and P. Mediterranea: A Review"

_bioengineering, 2019, doi:10.3390/bioengineering6040105_

Round 1

Reviewer 1 Report

The review is well organized and clearly written and I agree for publication. Please correct only a few typos as follows:

1. Lines 112-114: Look at the sentence: “At the early stationary phase (48 hrs.) P. mediterranea 9.1 yielded 6.9 g/L of partially purified EPS, 17-fold higher than in Pco A1 (3.9 g/L). PHA production was slightly higher in Pco A1 than in Pme 9.1 (respectively 0.92 g/L and 0.52 g/L) [37].”. The ratio between Pme 9.1 and Pco A1 EPS yields is 1.7; moreover, the ratio between Pco A1 and Pme 9.1 PHA yields is 1.7 too.

2. Lines 209-210: In the sentence: “Genomic studies which investigated potential correlations between the phenotype and genotype of Pco 388, Pco A1 and Pme 9.1 has shown that...” replace “has” with “have”

3. Lines 260-261: In the sentence:” The transcriptional profiles of Pco A1 and Pme 9.1 growing on a substrate with 2% of high-grade glycerol under inorganic nutrient-limited conditions was investigated” replace “was” with “were”

4. Line 273: “17-fold higher production of EPS (6.9 g/L compared to 3.9 g/L).”. Look at the comment 1

Author Response

The authors are grateful for your kind revision and suggestions and have accepted all of them being aware of their contribution to a better understanding of the review.

The review is well organized and clearly written and I agree for publication. Please correct only a few typos as follows:

Lines 112-114: Look at the sentence: “At the early stationary phase (48 hrs.) mediterranea 9.1 yielded 6.9 g/L of partially purified EPS, 17-fold higher than in Pco A1 (3.9 g/L). PHA production was slightly higher in Pco A1 than in Pme 9.1 (respectively 0.92 g/L and 0.52 g/L) [37].”. The ratio between Pme 9.1 and Pco A1 EPS yields is 1.7; moreover, the ratio between Pco A1 and Pme 9.1 PHA yields is 1.7 too.

In fact, the quantity of EPS in Pco A.1 is not 3.9 g/L, we have typed it wrongly. We made the correction 0.39 g/L. so that 17-fold higher is correct. Thank you so much!

Lines 209-210: In the sentence: “Genomic studies which investigated potential correlations between the phenotype and genotype of Pco 388, Pco A1 and Pme 9.1 has shown that...” replace “has” with “have”.

Done, thank you. 

Lines 260-261: In the sentence:” The transcriptional profiles of Pco A1 and Pme 9.1 growing on a substrate with 2% of high-grade glycerol under inorganic nutrient-limited conditions was investigated” replace “was” with “were”.

Done, thank you

Line 273: “17-fold higher production of EPS (6.9 g/L compared to 3.9 g/L).”. Look at the comment 1.

Corrected: 0.39 g/L

Reviewer 2 Report

The review provides interesting insights related with the polyhydroxyalkanoates synthesis using two Pseudomonas species: P. corrugata and P. mediterranea. The topic of this paper, related to biopolymers production, is very hot, and the information provided could be interesting for the scientific community and the readers of Bioengineering.

The manuscript presents comprehensive overview on metabolic  potential of P. corrugata and P. mediterranea for poluhydroxyalkanoates (PHA) synthesis. Manuscript  has the potential to provide interesting knowledge  about PHA production using pseudomonads, however before publication it needs improvement.

Main problem and suggestions:

Authors should remove form the Abstract the information which are results presented in scientific papers e.g. information about difference in gene expression and results of genes transfer (lines 20-23) It would be better to remove information about molecular mass of filmable PHA (line 17), otherwise you should provide information about molecular mass of other PHA. Cultivation of bacteria toward PHA synthesis in flasks and bioreactor could give absolutely different results therefore it is important to provide information about that (Chapter 2.) Authors should specify what does it mean that "P. corrugata A.1 produced 1.8 g/L of PHA with 51.5% in 72 h" It means that this strain accumulated highest amount of PHA in 72nd hour of cultivation or in cultivation provided for 72 h? (line 92). This same in line 98 and Table 1. Change "Recovery" into "recovery" (line 122) What is Time in Table 2? Is it time of cultivation. If yes, what is relation between time of cultivation and characteristics of PHA? Information about Transcriptional regulation of PHA synthesis could be controversial. Especially, fragment related to phaIF operon controlling by PC1. In mentioned work, presence of rho independent terminators was not proved. Additionally, action of phaD as transcriptional activator of PI and PC1 is rather speculative. Expression "polyhydroxyalkanoates" should be unified (e.g. line 13, 28 and 309).

Author Response

The authors are grateful for your kind revision and suggestions and have accepted all of them being aware of their contribution to a better understanding of the review.

The review provides interesting insights related with the polyhydroxyalkanoates synthesis using two Pseudomonas species: P. corrugata and P. mediterranea. The topic of this paper, related to biopolymers production, is very hot, and the information provided could be interesting for the scientific community and the readers of Bioengineering.

The manuscript presents comprehensive overview on metabolic  potential of P. corrugata and P. mediterranea for poluhydroxyalkanoates (PHA) synthesis. Manuscript  has the potential to provide interesting knowledge  about PHA production using pseudomonads, however before publication it needs improvement.

Main problem and suggestions:

Authors should remove form the Abstract the information which are results presented in scientific papers e.g. information about difference in gene expression and results of genes transfer (lines 20-23).

We agree and deleted.

It would be better to remove information about molecular mass of filmable PHA (line 17), otherwise you should provide information about molecular mass of other PHA.

We agree and deleted.        

Cultivation of bacteria toward PHA synthesis in flasks and bioreactor could give absolutely different results therefore it is important to provide information about that (Chapter 2.) Absolutely true. In our trials we have observed around a tenfold increase: 2.9 g/L vs. 26 g/L as specified in the revision

Authors should specify what does it mean that "P. corrugata A.1 produced 1.8 g/L of PHA with 51.5% in 72 h" It means that this strain accumulated highest amount of PHA in 72nd hour of cultivation or in cultivation provided for 72 h? (line 92). This same in line 98 and Table 1. No, time is referred to the amount of PHA at the end of 72 hour of cultivation.

Change "Recovery" into "recovery" (line 122) Done

What is Time in Table 2? Is it time of cultivation. If yes, what is relation between time of cultivation and characteristics of PHA?  Hard to say. Unfortunately, the only data we have are visual observations at the end of each experiment. We think they would not be helpful for the reader.   

Information about Transcriptional regulation of PHA synthesis could be controversial. Especially, fragment related to phaIF operon controlling by PC1. In mentioned work, presence of rho independent terminators was not proved. Additionally, action of phaD as transcriptional activator of PI and PC1 is rather speculative. We have rephrased the sentences regarding transcriptional study as follows: "In turn, PI and PC1 are controlled by PhaD which acts as a transcriptional activator as shown by the reduced promoter activity in the phaD- mutant [55]. Similar results were observed in P. putida KT2442 [23, 56]." About the presence of rho terminators they have been detected during genome annotation by sequence analysis, as described in the cited paper.

Expression "polyhydroxyalkanoates" should be unified (e.g. line 13, 28 and 309). The differences observed are the result of the various indications we have received any time we have submitted a paper. This time we unified as “polyhydroxyalkanoates”

Reviewer 3 Report

The here presented review it is comprenhensive, well-written and detailed. I recommend this review for publications if a minor number of issues are addressed:

Major comments:

The figures quality must be enhanced. The resolution of some of them is very low.

Minor comments:

Line 33: There are three categories of PHAs based on the unit repetition: scl, mcl and lcl.

Line 112: a period is missing

Line 127: redundant to write “process” twice in the same sentence

Line 211: Please briefly discuss about the PHA synthase existing types

Line 248-249: Please briefly argue about the different sizes of the PHA granules since some literature has already described this phenomenon.

Author Response

The authors are grateful for your kind revision and suggestions and have accepted all of them being aware of their contribution to a better understanding of the review.

The here presented review it is comprenhensive, well-written and detailed. I recommend this review for publications if a minor number of issues are addressed:

Major comments:

The figures quality must be enhanced. The resolution of some of them is very low.

We attempted to improve the quality and acting on the balance lux and contrast. We feel now are somewhat better. We have inserted an improved Fig 1.

Minor comments:

Line 33: There are three categories of PHAs based on the unit repetition: scl, mcl and lcl.

We know the categories of PHA are three, but many authors say they are mainly two. To be more precise we added this sentence: Less common and least studied are long chain length (lcl) PHAs, constituted by monomers with more than 14 carbon atoms

Line 112: a period is missing. Done

Line 127: redundant to write “process” twice in the same sentence.

We corrected in “cultivation” and “process”

Line 211: Please briefly discuss about the PHA synthase existing types.

We have added: This genetic system (class II) allows the Pseudomonas  to utilize medium-chain-length (mcl) monomers (C6–C14), whereas class I, III and IV systems polymerize short-chain-length (scl) monomers (C3–C5) [6].  

Line 248-249: Please briefly argue about the different sizes of the PHA granules since some literature has already described this phenomenon.

In our understanding the granule formation and distribution is not clear and more work is needed to find out a clear resolution. Moreover the theme is behind the focus of the review. Accepting your suggestion, we added a short sentence at the end of the phrase. “In addition, the wild type strain produced only a few large PHA inclusion bodies when grown with oleic acid, whereas the mutants showed numerous smaller PHA granules that line the periphery of the cells, as result of phasin activities”